# Why Did ZIKV Perinatal Outcomes Differ in Distinct Regions of Brazil? An Exploratory Study of Two Cohorts

**DOI:** 10.3390/v13050736

**Published:** 2021-04-23

**Authors:** Luana Damasceno, Ana Carolina B. Terzian, Trevon Fuller, Cassia F. Estofolete, Adriana Andrade, Erna G. Kroon, Andrea A. Zin, Zilton Vasconcelos, Jose P. Pereira, Márcia C. Castilho, Isa Cristina R. Piaulino, Nikos Vasilakis, Maria E. Moreira, Karin Nielsen-Saines, Flor E. Martinez Espinosa, Maurício L. Nogueira, Patricia Brasil

**Affiliations:** 1Acute Febrile Illnesses Laboratory, National Institute of Infectious Diseases, Oswaldo Cruz Foundation (OCRUZ), Rio de Janeiro 21040-900, RJ, Brazil; luana.damasceno@ini.fiocruz.br (L.D.); trevon.fuller@ini.fiocruz.br (T.F.); 2René Rachou Institute, Oswaldo Cruz Foundation (FIOCRUZ/Minas), Belo Horizonte 30190-002, MG, Brazil; anacarolinaterzian@gmail.com; 3Laboratory of Virology, School of Medicine (FAMERP), São José do Rio Preto 15090-000, SP, Brazil; cassiafestofolete@gmail.com; 4Institute of Biological Sciences, Federal University of Minas Gerais (UFMG), Pampulha-Belo Horizonte 31270-901, MG, Brazil; driquinha.drade@gmail.com (A.A.); kroone@icb.ufmg.br (E.G.K.); 5Fernandes Figueira Institute, Oswaldo Cruz Foundation (FIOCRUZ), Rio de Janeiro 22250-020, RJ, Brazil; andreazin@iff.fiocruz.br (A.A.Z.); zilton.vasconcelos@iff.fiocruz.br (Z.V.); josepaulo@globo.com (J.P.P.J.); bebethiff@gmail.com (M.E.M.); 6Department of Virology, Dr. Heitor Vieira Dourado Tropical Medicine Foundation, Manaus 69040-000, AM, Brazil; mcastilho@fmt.am.gov.br; 7Postgraduate Program in Tropical Medicine, Amazonas State University, Manaus 69040-000, AM, Brazil; isa.crissy@gmail.com; 8Department of Pathology, University of Texas Medical Branch, Galveston, TX 77555, USA; nivasila@utmb.edu; 9Department of Pediatrics, David Geffen School of Medicine, University of California, Los Angeles, CA 90095, USA; knielsen@mednet.ucla.edu; 10Laboratory of Territory, Environment, Health, and Sustainability, Leônidas & Maria Deane Institute, Oswaldo Cruz Foundation (FIOCRUZ/Amazonia), Manaus 69057-070, AM, Brazil; flor.espinosa@fiocruz.br; 11Department of Malaria, Dr. Heitor Vieira Dourado Tropical Medicine Foundation, Manaus 69040-000, AM, Brazil

**Keywords:** Zika, pregnancy, obstetrics, arboviruses, dengue

## Abstract

The Zika virus (ZIKV) epidemic in Brazil occurred in regions where dengue viruses (DENV) are historically endemic. We investigated the differences in adverse pregnancy/infant outcomes in two cohorts comprising 114 pregnant women with PCR-confirmed ZIKV infection in Rio de Janeiro, Southeastern Brazil (*n* = 50) and Manaus, in the north region of the country (*n* = 64). Prior exposure to DENV was evaluated through plaque reduction neutralizing antibody assays (PRNT 80) and DENV IgG serologies. Potential associations between pregnancy outcomes and Zika attack rates in the two cities were explored. Overall, 31 women (27%) had adverse pregnancy/infant outcomes, 27 in Rio (54%) and 4 in Manaus (6%), *p* < 0.001. This included 4 pregnancy losses (13%) and 27 infants with abnormalities at birth (24%). A total of 93 women (82%) had evidence of prior DENV exposure, 45 in Rio (90%) and 48 in Manaus (75%). Zika attack rates differed; the rate in Rio was 10.28 cases/10,000 and in Manaus, 0.6 cases/10,000, *p* < 0.001. Only Zika attack rates (Odds Ratio: 17.6, 95% Confidence Interval 5.6–55.9, *p* < 0.001) and infection in the first trimester of pregnancy (OR: 4.26, 95% CI 1.4–12.9, *p* = 0.011) were associated with adverse pregnancy and infant outcomes. Pre-existing immunity to DENV was not associated with outcomes (normal or abnormal) in patients with ZIKV infection during pregnancy.

## 1. Introduction

In recent decades, Brazil has experienced significant flavivirus epidemics, including dengue serotype 1 (DENV-1) in 1998, DENV-3 in 2002, DENV-2 in 2008, DENV-4 in 2010, Zika virus (ZIKV) in 2015–2016, and the reemergence of the Yellow Fever (YF) in 2017 [1,2]. Among these flaviviruses, dengue and Zika had the highest prevalence and the greatest degree of geographic overlap and co-circulation in large cities [2,3].

The morbidity associated with ZIKV infection in pregnancy was first noted when an unprecedented increase in microcephaly was reported in the months following the Zika epidemic, initially in northeastern Brazil, and subsequently spreading to other parts of the country. Not long thereafter, an association between the two epidemics (Zika and microcephaly) was confirmed [4,5,6]. In November 2015, with thousands of confirmed cases of the disease all over the world, the World Health Organization (WHO) declared an international public health emergency [7,8,9].

Differing incidence rates for Zika have been reported in regions of Brazil where DENV has historically been endemic or hyperendemic. Thus, most individuals who contracted ZIKV infection likely had prior exposure to at least one DENV serotype [9,10]. In 2016, the Brazilian Ministry of Health estimated that the incidence of Zika in the state of Rio de Janeiro, in the southeastern region of the country, was notably higher than that in the state of Amazonas, in the northern region of Brazil, at 430/100,000 inhabitants and 112/100,000, respectively [11]. Similarly, during the same period, the proportion of cases of microcephaly was also different in these two regions. Between 2015 and 2016, Rio de Janeiro reported 578 cases of microcephaly, while Amazonas reported 25 cases [12].

These distinct scenarios lead to the question of whether an individual’s history of DENV exposure could influence the immune response to ZIKV, as these two viruses are antigenically related. While the antigenic structure of ZIKV resembles that of other pathogenic flaviviruses, such as YF, West Nile, and Japanese Encephalitis, ZIKV is most similar to DENV [13]. The two viruses share a conserved surface glycoprotein envelope (E), which is involved in antibody recognition and humoral cross-reactivity between the viruses, producing responses that can either neutralize the virus or have immunopathogenic implications, a phenomenon referred to as antibody-dependent enhancement (ADE) [14,15,16].

To date, the real influence of preexisting flavivirus immunity in pregnant women with ZIKV infection and adverse outcomes has not been entirely elucidated [10,17] in regions where other flaviviruses co-circulate. The reduction in the incidence of Zika cases since 2016 has made it more difficult to conduct prospective studies that would provide a better understanding of the influence, if any, of prior immunity to DENV on the clinical course of ZIKV infection [18,19]. There is evidence from in vitro studies that lower DENV antibody titers are associated with increased viral replication of ZIKV, via the ADE phenomenon, which could potentially result in a higher risk of adverse neonatal outcomes [15,20]. On the other hand, high DENV antibody titers could reduce susceptibility to Zika and limit its pathogenicity [17,21,22]. It remains unknown whether human populations actually develop this type of cross-immunity and if it affects gestational outcomes [16,23], adverse events, or perinatal transmission [20,24,25].

To investigate the differences in rates of adverse outcomes between these two regions of Brazil we examined the relationship between pre-existing immunity to DENV and abnormalities in symptomatic pregnant women who had RT-PCR-confirmed ZIKV infection in two cohorts from Rio de Janeiro and Manaus. We also looked for a potential association between Zika attack rates during the epidemic and the rate of adverse outcomes in the two cohorts. This question has significant public health implications since most studies investigating these interactions have been in animal models and there is a paucity of human studies addressing this topic.

## 2. Materials and Methods

### 2.1. Clinical Samples

We analyzed 114 serum specimens collected from pregnant women with acute ZIKV infection between September 2015 and June 2016. Participants were recruited at two sites: the Heitor Vieira Dourado Foundation for Tropical Medicine (FMT-HVD), a referral center for infectious diseases in Manaus (*n* = 64), and the acute febrile illness clinic at the Oswaldo Cruz Foundation (FIOCRUZ) in Rio de Janeiro (*n* = 50). Details of the study design and data collection for these cohorts have been previously described [26,27,28]. 

### 2.2. Adverse Outcomes

We considered the presence of the following adverse outcomes or signs: fetal death/stillbirth (*n* = 4); severe microcephaly at birth, defined as three standard deviations below normal head circumference [29] (*n* = 6); clinical neurologic findings (seizures, severe irritability, dysphagia, hypertonia, and hyperreflexia) (*n* = 10); structural brain abnormalities (intracranial subcortical calcifications or ventriculomegaly, based on CT scans) (*n* = 10); congenital contractures, such as clubfoot or arthrogryposis (*n* = 1); smallness for gestational age (*n* = 4); abnormalities based on retinal fundus imaging performed using a fiber optic camera (RetCam) (*n* = 5); hearing deficits diagnosed by brainstem evoked response audiometry (BERA) [30,31] (*n* = 6). Women infected with CMV, toxoplasma, rubella, HSV, and CHIKV during pregnancy were excluded from the study.

### 2.3. Laboratory Diagnosis of ZIKV and DENV Infection

ZIKV infection was confirmed by RT-qPCR of blood and/or urine specimens [32]. Prior immunity to dengue was assessed via serological tests for anti-DENV IgG (with ELISA). In addition, to minimize the effect of potential cross-reactivity between DENV serotypes and ZIKV antibodies, maternal serotype-specific immunity to DENV was evaluated by plaque reduction neutralization tests (PRNT 80) following a standardized protocol, as discussed below.

We defined prior DENV exposure as IgG positive or PRNT 80 positive (≥1:20). The maternal immune response pattern was classified as monotypic (the presence of neutralizing antibodies for a single DENV serotype) or multitypic (defined as the presence of neutralizing antibodies for more than one DENV serotype). Individuals whose serology results for IgG were negative and had no neutralizing antibodies based on the PRNT 80 were classified as having had no prior DENV exposure.

#### 2.3.1. DENV Virus Stock

The cell line *Ae. albopictus* C6/36 was maintained in Leibovitz-15 medium (L-15, Cultilab, Campinas, SP, Brazil) at 28 °C. Growth media were supplemented with 10% Fetal Bovine Serum (FBS) (Cultilab, Campinas, SP, Brazil; GIBCO, Waltham, MA, USA), 100 U/mL of penicillin, and 100 μg/mL of streptomycin (GIBCO, Waltham, MA, USA). For the assay, the virus strains used were DENV-1 Mochizuki; DENV-2 NGC; DENV-3 (patient isolate from São Paulo, Brazil); DENV-4 Pe/0081 (patient isolate from Pernambuco, Brazil) and propagated as previously described [33].

The supernatant was harvested 5 days (DENV-2 and 4) or 7 days (DENV-1 and 3) post-infection (dpi) and viral stocks were titered by plaque assay in Vero E6 cells. The virus inoculum was removed following the addition of a semi-solid medium (Minimum Essential Medium (MEM) 1×, 1% FBS, 1.5% carboxymethylcellulose). At 7 dpi, the plates were fixed with cold 1:1 methanol/acetone, and the foci were stained immunologically with mouse anti-DENV-2 hyper-immune ascites fluid (MIAF) (1:3000), as previously described [34,35,36]. The results were expressed as focus-forming units per milliliter (FFU/mL).

#### 2.3.2. Plaque Reduction Neutralization Tests

The Vero E6 cell line was maintained in Eagle’s Minimum Essential Medium (MEM, Cultilab, Campinas, SP, Brazil) at 37 °C in a humidified atmosphere containing 5% CO2. Growth media were supplemented with 10% Fetal Bovine Serum (FBS) (Cultilab, Campinas, SP, Brazil; GIBCO, Waltham, MA, USA), 100 U/mL of penicillin, and 100 μg/mL of streptomycin (GIBCO, Waltham, MA, USA). Sera samples were inactivated at 56 °C for 60 min and assayed to determine the specific neutralization antibody titers, as described previously [34]. Briefly, the PRNT was performed in 24-well plates of Vero cell culture using a fixed virus inoculum (approximately 1000 FFUs) against varying serum dilutions (1:20–1:640). The plates were then fixed with cold 1:1 methanol/acetone and the foci were stained immunologically with mouse anti-DENV-2 hyper-immune ascites fluid (MIAF) (1:2000), as previously described [26,27]. The PRNT titers were scored as reciprocal of the highest dilution of serum that inhibited 80% of the plaques (PRNT 80). Samples scored as PRNT 80 < 20 were considered negative.

#### 2.3.3. DENV IgG ELISA

DENV IgG was detected using the Panbio^®^ Dengue IgG Indirect ELISA (Abbott Laboratories, Chicago, IL, USA), according to manufacturer instructions. The assay did not discriminate between DENV serotypes. The plate was read at 450 nm using a Spectramax Plus ELISA reader (Molecular Devices, LLC, San Jose, CA, USA).

#### 2.3.4. Avidity Assay

Serum avidity was measured using a modified protocol adapted from the Panbio^®^ Dengue IgG Indirect ELISA (Abbott Laboratories, Chicago, IL, USA), with guanidine hydrochloride (GuHCl) incubation [4,5]. We opted for this kit due to its high sensitivity and specificity for the four DENV serotypes. The optimal concentration to disrupt antigen binding to low avidity antibodies, reducing the antibody binding to 50%, was 2.0 M GuHCl. A control serum with intermediate avidity antibodies was used as a control. Diluted duplicate serum samples were incubated for 30 min at 37 °C and washed, and each replicate was incubated with 100 μL of 2.0 M GuHCl or phosphate-buffered saline with 1% BSA and 0.2% Tween20 (PBS 1×). After 15 min of incubation, at 37 °C, the wells were washed four times and the assay was completed according to the manufacturer’s instructions. The plate was read at 450 nm using a Spectramax Plus ELISA reader (Molecular Devices, LLC, San Jose, CA, USA). PBS 1× was used in the background control wells and sera from DENV-naïve individuals were used as negative controls. Avidity was calculated as a percentage of the optical density (O.D.) treated with PBS and compared to the O.D. remaining after treatment with the GuHCl (% = (GuHCL O.D./PBS O.D.) * 100), as described previously [37].

### 2.4. Attack Rates

The Zika attack rate in each city was defined as the number of cases reported during the study period divided by the total population size. Zika cases were obtained from the Notifiable Diseases Information System of the Ministry of Health of Brazil (Sistema de Informação de Agravos de Notificação (SINAN). The population of each city was estimated from the national census (IBGE 2016).

### 2.5. Statistical Analyses

The rate of prior DENV exposure among pregnancies with adverse outcomes was compared to the rate in pregnancies with normal outcomes using a Chi-squared test. In addition, we used a regression model to assess the relationship between adverse gestational outcomes and variables, such as the mother’s chronological age, gestational age at delivery, and anti-DENV titers by PRNT 80 for the four serotypes. The model also accounted for the nesting of the participants within cities. Failure to address the clustering of the participants by city would lead to an overestimation of the degree of freedom of the data, which could result in erroneous conclusions about the significance of a variable addressing adverse outcomes. To account for such correlation, we utilized a generalized linear mixed model [38], in which the dependent variable was the presence or absence of gestational adverse events. The aforementioned maternal variables were fixed effects, and the city was a random effect with the city-specific attack rate as a covariate. The parameters of the model were estimated using the generalized linear mixed-effects model (glmer) procedure in R 3.5.1 software. The significance of the independent variables was assessed via Wald’s test.

The study was approved by the local Institutional Review Board (IRB). The authors vouch for the accuracy and completeness of the data and analyses, and for the fidelity of the study to the protocol. Written informed consent was obtained from all participants.

## 3. Results

In the samples selected from the Rio de Janeiro cohort, the rate of adverse outcomes in that group of patients was 54%, while the adverse outcome rate for patients who had specimens in the Manaus cohort was 6%, similar to the results reported for the original cohorts [26,28]. The frequency of prior exposure to DENV was 90% in Rio de Janeiro and 75% in Manaus (Table 1).

ELISA assays were performed on 114 specimens with the following results: 93 participants (82%) had prior DENV exposure (PRNT ≥ 20 or ELISA-positive) and 21 (18%) had negative DENV antibody results (PRNT < 20 and ELISA-negative or indeterminate) (Table 2). Of the previously exposed DENV pregnant women (PW), four babies (4.3%) died before birth, 24 had abnormal findings (25.8%), and 65 (69.9%) had normal outcomes. Among the 21 PW with no previous exposure to DENV (IgG-negative and PRNT < 1:20), 3 had infants with abnormal outcomes (14.3%) and 18 infants (85.7%) were normal (*p* = 0.35) (Table 2).

Of the 114 samples tested by PRNT, the most prevalent serotypes were DENV-1 (68.4%), DENV-2 (59.7%), and DENV-4 (42.98%), and the least prevalent serotype was DENV-3 (4.4%). Multitypic DENV exposure had occurred in 89% (*n* = 57) of the participants, based on the neutralization of more than one DENV serotype (Figure 1).

The median log2 titer for specific serotypes in pregnancies with abnormal outcomes was 6.32 DENV-1 (95% CI 0–9.32), 5.32 DENV-2 (95% CI 0–9.32), 0 DENV-3 (CI 0–0), and 4.91 DENV-4 (CI 0–6.32). For normal outcomes, the median log2 titer for specific serotypes was 5.9 DENV-1 (95% CI 0–8.3), 4.3 DENV-2 (95%CI 0–7.1), 0 DENV-3 (CI 0–0), and 0 DENV-4 (CI 0–4.3) (Table 3 and Figure 1). Fully, 70% of pregnancies with adverse outcomes had neutralizing antibodies against at least two DENV serotypes. However, the association between monotypic or heterotypic DENV infection and adverse gestational outcomes was not statistically significant (Table 4).

The avidity assay was performed on 96 samples. There was no difference in DENV avidity between mothers with adverse gestational outcomes and those with normal outcomes (Appendix A). No effect of maternal DENV titer during pregnancy and abnormal outcomes were observed. However, there was a significant association between the trimester of infection (*p* = 0.027) and the Zika attack rate in Rio de Janeiro (10.28 cases/10,000) and in Manaus (0.6 cases/10,000) (*p* < 0.001) (Table 4). In the regression model, the coefficient of the attack rate was a positive number, indicating that the probability of an adverse outcome increased with the increasing incidence of ZIKV infection (Table 4 and Figure 2).

## 4. Discussion

In the present study, only the Zika attack rate and the time of infection during pregnancy were significantly associated with the risk of adverse outcomes. Women infected in the first trimester were more likely to deliver infants with abnormalities, adding to the growing evidence that ZIKV infection early in pregnancy has a generalizable association with microcephaly [29]. Our findings regarding Zika attack rates support previous studies that demonstrated geographic differences in Zika incidence, such as the disproportionately high rates of Zika and microcephaly in northeastern Brazil [39,40] and a lower rate of microcephaly identified in Manaus.

However, these results should be interpreted with caution. The real proportion of ZIKV infection in the population in many regions of Brazil may not be represented because of missing data in the notification system SINAN, prior to the establishment of a nationwide ZIKV surveillance system [19,41].

A theory has also been mooted that pre-existing immunity to co-circulating flaviviruses from prior natural infection (e.g., dengue, YF) or from vaccination to YF may influence ZIKV disease severity and pregnancy outcomes [42]. As YF vaccination is compulsory in the Amazon region, it could protect the population from ZIKV infection, as demonstrated in a Brazilian study based on population spatial–temporal clusters between YF vaccination coverage and ZIKV-associated microcephaly [43]. The offspring of pregnant women in regions with high YF vaccination coverage seemed to experience less risk of microcephaly and overall congenital Zika syndrome. This association suggests that flavivirus antibodies could be protective against severe manifestations of ZIKV infection, including during pregnancy [43]. This observation could also be due to a lower Zika attack rate conferred by YF immunization, translating into a smaller number of cases of microcephaly, which generally occur in 3 to 5% of infants exposed to ZIKV in utero [26,44].

We did not detect any temporal associations between prior DENV infection and adverse pregnancy outcomes, as measured by the IgG ELISA test avidity in either Rio de Janeiro or Manaus (Table 4). This is distinct from a recent ecological study that demonstrated that the risk of microcephaly was lower in municipalities that experienced a DENV epidemic up to six years prior to the introduction of ZIKV into the community [18].

In our results, pre-existing immunity to DENV, neutralizing antibody (NAbs) titers, and multitypic DENV immunity during pregnancy neither conferred protection nor increased the risk of adverse outcomes. These are compatible with previous studies in non-human primates [45] but unlike prior studies in humans [21,46]. This difference may be attributable to differences in the study design, the definition of ZIKV infection, and the ascertainment of outcomes. Furthermore, unlike the present analysis, previous studies did not utilize sera from ZIKV-infected participants diagnosed via RT-PCR.

The strengths of our study include a careful classification of infant outcomes, made possible through detailed assessments at birth by a multidisciplinary team. This type of ascertainment is not possible in secondary analyses performed through routinely collected surveillance data., In addition, the use of a highly sensitive and specific PRNT assay to characterize pre-existing DENV immunity and the use of sera collected during the acute phase of ZIKV RT-PCR-confirmed infection lends credibility to our results. Our study design also allowed us to determine the trimester of maternal ZIKV infection and to prospectively evaluate fetal losses and abnormalities before birth.

Our main limitations are the modest sample size and convenience sample selection. For this reason, the extent to which our findings can be generalized to other populations remains unknown. Our population consisted of women with symptomatic ZIKV infection, which is what allowed us to determine the timing of infection. Furthermore, since we classified neonates as normal or abnormal at delivery, we may have missed adverse outcomes that emerge later in development [44].

In summary, in this study, women with prior DENV exposure who were infected with ZIKV during pregnancy did not experience a higher rate of adverse outcomes at delivery in these two regions. However, disentangling the effects of these infections on gestation remains challenging due to the similar clinical presentations of DENV and ZIKV illnesses and the waning of DENV antibody titers over time.

## Figures and Tables

**Figure 1 viruses-13-00736-f001:**
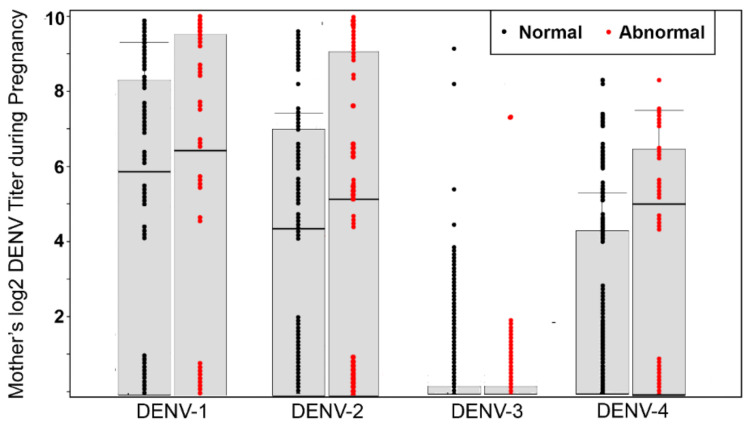
Effect of maternal DENV titer during pregnancy on adverse outcomes (*N* = 114). The rate of abnormal births in mothers with DENV-neutralizing antibodies was different than that of mothers lacking such antibodies.

**Figure 2 viruses-13-00736-f002:**
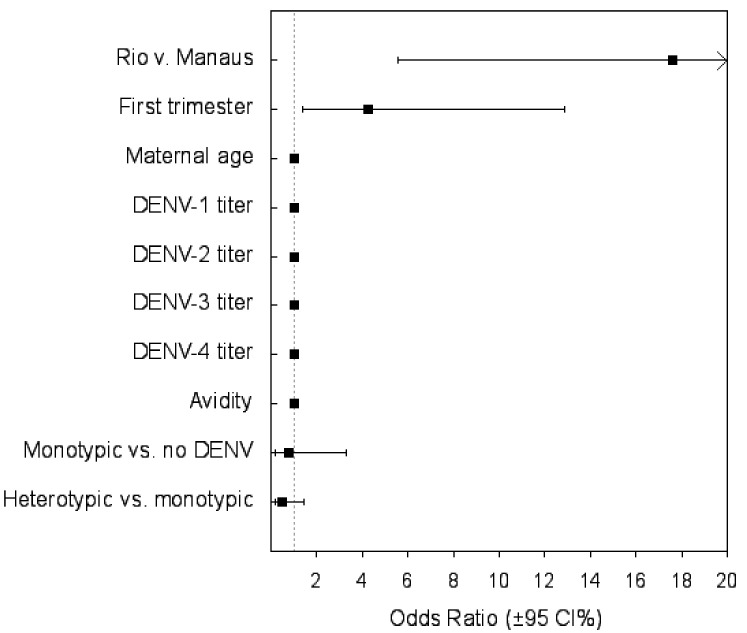
Risk factors for infant abnormalities at delivery (*N* = 114).

**Table 1 viruses-13-00736-t001:** Maternal characteristics and infant outcomes stratified by municipality.

	Municipality			*p*
Maternal Characteristics	Total	Rio de Janeiro, RJ (*N* = 50)	Manaus, AM (*N* = 64)	
**Age**	***N*** **(%)**	***N*** **(%)**	***N*** **(%)**	0.09
<20	16 (14)	5 (10)	11 (17)	
20–29	47 (41)	16 (32)	31 (48)	
30–39	45 (39)	26 (52)	19 (30)	
>40	6 (5)	3 (6)	3 (5)	
**Gestational age at the time of maternal ZIKV infection (weeks)**	***N*** **(%)**	***N*** **(%)**	***N*** **(%)**	0.004
<=26	75 (66)	40 (80)	35 (55)	
>=27	35 (31)	7 (14)	28 (44)	
Missing	7 (6)	3 (6)	4 (6)	
**DENV IgG**				0.05
Reactive	93 (81)	45 (90)	48 (75)	
Non-reactive	21 (18)	5 (10)	16 (25)	
**Adverse Outcomes**	31 (27)	27 (54)	4 (6)	<0.001
Disproportionate Microcephaly and cerebral calcifications, irritability, hypertonicity, abnormal funduscopic and hearing exam, and clubfoot or arthrogryposis	6 (19)	5 (16)	1 (3)	
Proportional microcephaly, hypertonicity, or abnormal neurologic exam	2 (6)	2 (6)	0	
Smallness for gestational age (SGA) and hypotonicity, or hearing deficits and hypertonicity; structural abnormality in neuroimaging	4 (13)	4 (13)	0	
Abnormal funduscopic exam and irritability, dysphagia, or hypertonicity and abnormal reflexes	4 (13)	4 (13)	0	
Hearing deficits, hypertonicity, and abnormal reflexes	3 (10)	3 (10)	0	
Hypertonicity and hyperreflexia; knee fovea, redundant scalp	4 (13)	4 (13)	0	
Dysphagia and irritability	1(3)	1 (3)	0	
Image structural abnormalities	3 (10)	3 (10)	0	
Fetal death/stillbirth	4 (13)	3 (10)	1 (3)	
Normal findings	83 (73)	23 (46)	60 (94)	

**Table 2 viruses-13-00736-t002:** Maternal characteristics and infant outcomes, stratified by dengue virus (DENV) exposure.

	DENV Exposure			
Maternal Characteristics	Total	Prior DENV Exposure (*N* = 93)	No Prior DENV Exposure (*N* = 21)	*p*
**Age**	***N*** **(%)**	***N*** **(%)**	***N*** **(%)**	0.179
<20	16 (14.04)	10 (10.8)	6 (28.6)	
20–29	47 (41.23)	41 (44.1)	6 (28.6)	
30–39	45 (39.47)	37 (39.8)	8 (38.1)	
>40	6 (5.26)	5 (5.4)	1 (4.8)	
**Gestational age at ZIKV infection with (weeks)**	***N*** **(%)**	***N*** **(%)**	***N*** **(%)**	0.603
<=26	75 (65.79)	62 (66.7)	13 (61.9)	
>=27	35 (30.7)	27 (29)	8 (38.1)	
Missing	8 (7.02)	4 (4.3)	4 (19)	
**Municipality**				0.052
Rio de Janeiro	50 (43.86)	45 (48.4)	5 (23.8)	
Manaus	64 (56.14)	48 (51.6)	16 (76.2)	
**Infant Outcomes**				0.181
Altered	31 (27.2)	28 (30.1)	3 (14.2)	
Non-altered	83 (70.18)	65 (69.9)	18 (85.8)	

**Table 3 viruses-13-00736-t003:** Maternal characteristics and neonatal outcomes.

	Infant Outcomes			
Maternal Characteristics	Total	Normal Findings	Abnormal Findings	*p*
**Age**	***N*** **(%)**	***N*** **(%)**	***N*** **(%)**	0.34
<20	16 (14)	10 (16)	6 (12)	
20–29	44 (38)	28 (44)	16 (31)	
30–39	48 (42)	22 (35)	26 (51)	
>40	6 (5)	3 (5)	3 (6)	
**Gestational age at the time of maternal ZIKV infection(weeks)**	***N*** **(%)**	***N*** **(%)**	***N*** **(%)**	<0.001
<=26	75 (66)	53 (66)	22 (65)	
>=27	35 (31)	27 (34)	8 (24)	
Missing	4 (4)	0	4 (12)	
**Log2 DENV PRNT 80 (*N* = 114, median, IQR)**				0.474
DENV-1	6.3 (0–8.1)	5.9 (0–8.3)	6.3 (0–9.3)	
DENV-2	4.4 (0–7.3)	4.3 (0–7.1)	5.3 (0–9.3)	
DENV-3	0	0	0	
DENV-4	0 (0–5.4)	0 (0–4.3)	4.9 (0–6.3)	
Municipality	***N*** ** (%)**	***N*** ** (%)**	***N*** ** (%)**	<0.001
Rio de Janeiro	50 (43.9)	23 (28.7)	27 (79.4)	
Manaus	64 (56.1)	57 (71.3)	7 (20.6)	

**Table 4 viruses-13-00736-t004:** Multivariate logistic regression of predictors of infant abnormalities at birth (*N* = 114).

Variables	Odds Ratio (95% CI)	*p*
Municipal-scale variable		
Attack rate Rio vs. Manaus	17.6 (5.55–55.88)	<0.001
Maternal categorical variables		
First trimester ZIKV infection	4.26 (1.4–12.9)	0.011
Monotypic DENV infection vs. no DENV infection	0.78 (0.18–3.31)	0.73
Heterotypic vs. monotypic DENV infection	0.466 (0.154–1.413)	0.18
Maternal continuous variables		
Maternal age	1.0 (0.99–1)	0.59
DENV-1 titer	0.99 (0.99–1)	0.198
DENV-2 titer	0.99 (0.99–1)	0.846
DENV-3 titer	0.99 (0.98–1.0)	0.32
DENV-4 titer	1 (0.99–1.01)	0.7
Test of avidity	0.99(0.96–1.02)	0.523

## Data Availability

De-identified data are available from the authors upon request.

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
