# Peer review of "Why Did ZIKV Perinatal Outcomes Differ in Distinct Regions of Brazil? An Exploratory Study of Two Cohorts"

_viruses, 2021, doi:10.3390/v13050736_

Round 1

Reviewer 1 Report

The manuscript aims to investigate the differences in the frequency of adverse outcomes in two cohorts of pregnant women with ZIKV in Rio de Janeiro (Southeastern) and Manaus (Northeastern) of Brazil in according of the pre-existing immunity to DENV. Also, they looked for a potential association between ZIKV attack rates during epidemic and rate of adverse outcomes in both cohorts. The cohorts have been well validated in other articles published by authors. In spite of the small cohort, the study is well conducted, and the results contributed to clarify if the previous immunity to DENV infections influences ZIKV perinatal outcomes.

Minor reviews:

Why the authors used the expression “ZIKV attack rate”? Is it the same ZIKV infection rate? So, is more appropriate to use ZIKV infection rate.

There are many tables and figures (4 tables and 4 figures). So, I suggest Figure 2 and Figure 3 as supplementary material or data report.

In Table 3, the authors should write DENV-“1”,DENV-“2”, DENV-“3” and DENV-“4”.

Women infected with CMV, Toxoplama, rubella and HSV during pregnancy were excluded from the study. Authors would debate about others virus frequent in the Brazil region as Chikungunya and the cohort studied.

Author Response

Point 1: Why the authors used the expression “ZIKV attack rate”? Is it the same ZIKV infection rate? So, is more appropriate to use ZIKV infection rate. 

Response 1:

Thanks for your astute observation. Indeed, we wanted to refer to the incidence of Zika disease (the number of people with Zika divided by the number of people exposed during the epidemic period). We used a wrong term – “ZIKV attack rate”. Instead, we changed it to “Zika attack rate”, to make it clearer.

Point 2: There are many tables and figures (4 tables and 4 figures). So, I suggest Figure 2 and Figure 3 as supplementary material or data report.

Response 2:

We placed Figures 2 and Figure 3 as supplementary material.

Point 3: In Table 3, the authors should write DENV-“1”,DENV-“2”, DENV-“3” and DENV-“4”. Women infected with CMV, Toxoplama, rubella and HSV during pregnancy were excluded from the study. Authors would debate about others virus frequent in the Brazil region as Chikungunya and the cohort studied.

Response 3:

We changed the way DENV was presented in the table 3. In the study all pregnant women had serologic and PCR tests negative for CHIKV and this information was included:  “Women infected with CMV, toxoplasma, rubella, HSV, and CHIKV during pregnancy were excluded from the study” (line 128). 

Reviewer 2 Report

Luana et al. have investigated the difference in the frequency of ZIKA infected subjects with or without prior exposure to DENV. Overall, there are no major changes needed.

I would suggest authors revise the title and abstract. Nevertheless, I am concerned with the abstract than the title. Try to add some more text to connect the abstract well. 

The authors have briefly highlighted the strengths and limitations of the study, which is appreciative. 

The manuscript is well written; however,  I would suggest authors have the manuscript proofread by a colleague expert in academic writing to polish it further and improve the sentence structure. 

Author Response

Point 1: I would suggest authors revise the title and abstract. Nevertheless, I am concerned with the abstract than the title. Try to add some more text to connect the abstract well. 

Response 1: Thanks for your observation. We rewrote the title and the abstract, and believe the abstract now better summarized the manuscript. 

Point 2: The manuscript is well written; however, I would suggest authors have the manuscript proofread by a colleague expert in academic writing to polish it further and improve the sentence structure. 

Response 2:

 The text was edited by someone experienced in English academic writing.